# The Neural Signatures of Shame, Embarrassment, and Guilt: A Voxel-Based Meta-Analysis on Functional Neuroimaging Studies

**DOI:** 10.3390/brainsci13040559

**Published:** 2023-03-26

**Authors:** Luca Piretti, Edoardo Pappaianni, Claudia Garbin, Raffaella Ida Rumiati, Remo Job, Alessandro Grecucci

**Affiliations:** 1Clinical and Affective Neuroscience Laboratory, Department of Psychology and Cognitive Sciences, University of Trento, 38068 Rovereto, Italy; 2Marica de Vincenzi Onlus Foundation, 38122 Trento, Italy; 3Mental Health Services in the Capital Region of Denmark, Psychiatric Center Ballerup, 2750 Ballerup, Denmark; 4Neuroscience and Society Laboratory, Neuroscience Area, SISSA, 34136 Trieste, Italy; 5Center for Medical Sciences, University of Trento, 38122 Trento, Italy

**Keywords:** self-conscious emotions, shame, embarrassment, guilt, moral emotions, anterior insula

## Abstract

Self-conscious emotions, such as shame and guilt, play a fundamental role in regulating moral behaviour and in promoting the welfare of society. Despite their relevance, the neural bases of these emotions are uncertain. In the present meta-analysis, we performed a systematic literature review in order to single out functional neuroimaging studies on healthy individuals specifically investigating the neural substrates of shame, embarrassment, and guilt. Seventeen studies investigating the neural correlates of shame/embarrassment and seventeen studies investigating guilt brain representation met our inclusion criteria. The analyses revealed that both guilt and shame/embarrassment were associated with the activation of the left anterior insula, involved in emotional awareness processing and arousal. Guilt-specific areas were located within the left temporo-parietal junction, which is thought to be involved in social cognitive processes. Moreover, specific activations for shame/embarrassment involved areas related to social pain (dorsal anterior cingulate and thalamus) and behavioural inhibition (premotor cortex) networks. This pattern of results might reflect the distinct action tendencies associated with the two emotions.

## 1. Introduction

Moral emotions are crucial in regulating social interactions, as they promote the welfare of society or of other people [1] Indeed, they provide the emotional drive to properly behave in social interactions [2,3,4], forcing individuals to implement strategies that are optimal over a long period, even though they do not appear functional in the short period [5,6].

It has been proposed that moral cognition depends on the prefrontal, temporal, and limbic circuits, and is associated with the integration of context-independent and -dependent information and with the accompanying emotional reactions (event-feature-emotion complexes model, EFEC) [7,8]. Specifically, the prefrontal cortex seems to be responsible for representing context-dependent knowledge of event sequences [9,10], the temporal lobes for perceiving social cues and for representing context-independent social semantic knowledge [11,12,13], and the limbic system for the generation of emotional and motivational states [14]. Hence, according to the EFEC model, the generation of moral emotions, including self-conscious emotions, relies on the integrity of a network including prefrontal, temporal, and limbic areas [7,8].

Several studies investigating the neural substrates of moral cognition [15,16] confirmed the anatomical predictions of this model and better defined the topography of the brain areas associated with moral processing. Indeed, they showed that the ventromedial and dorsomedial prefrontal cortices (vmPFC and dmPFC, respectively), the temporo-parietal junction (TPJ), the precuneus, the posterior cingulate cortex, the left amygdala, the anterior temporal lobes (ATL), and the lateral orbitofrontal cortex were consistently found activated in neuroimaging studies investigating moral processing [3,4,15,16].

Among moral emotions, a sub-group of emotions (e.g., shame, embarrassment, guilt, and pride) defined as self-conscious emotions helps individuals to navigate the complexities of fitting into groups [1], satisfying the human need of belonging to social groups [17]. Self-conscious emotions are evoked by self-reflection and self-evaluation [18], and occur when social norms or agreed-upon social rules are violated [19], providing an immediate feedback that promotes inhibition or reinforcement of behaviour [3,4,18] (see Table 1). One case in point is shame, which has been proposed as an algorithm the brain uses to inhibit socially and morally unwanted behaviours [3,4].

While the EFEC model might explain the cognitive processes underlying all self-conscious emotions, which are all induced by moral and social norm violation [19], it does not make any prediction about the different processes that might occur in different types of emotions such as the negative self-conscious emotions. Indeed, even though shame, embarrassment, and guilt are often (but culpably) used interchangeably, they appear to be substantially different [20]. Shame is typically elicited by the belief that the individual’s violation of standards of morality, aesthetics, or competence defines who the individual is [21]. Hence, it involves the way the individual perceives themselves and how they believe other people see them and their inadequacy to fulfil social standards [22]. The distinction between shame and embarrassment is still a matter of debate (for review see [23]). If, on the one hand, embarrassment might be considered a dimension of shame [24], on the other, it might represent a distinct emotional entity [25,26]. Embarrassment seems related to trivial social transgressions, occurring suddenly and in public contexts, especially in the presence of individuals with equal or higher hierarchical social status [1,26,27,28,29]. Conversely, shame emerges when one personally perceives the serious violation of a moral norm, which might be also experienced in private situations [26,29]. Furthermore, shame and embarrassment also differ in intensity (i.e., shame is more intense than embarrassment) [30], in duration (i.e., shame is more persistent than embarrassment) [31], and in the focus of attention (i.e., shame affects the self, embarrassment affects the persona, the apparent self). However, these two emotions also have some features in common. They are associated with the same specific physiological reactions (e.g., blushing) [32] and the same action tendency, leading people to hide and reduce their social presence, making movement and speech more difficult and less likely [25,26,33,34]. However, it has also been reported that, differently from shame, embarrassment leads to reparative behaviours to re-gain social approval [25,35,36]. At the neural level, shame has been selectively associated with the dorso-lateral prefrontal cortex (dlPFC), the posterior cingulate cortex, and the sensory-motor cortex, whereas embarrassment has been associated with the ventro-lateral prefrontal cortex (vlPFC), the amygdala, and occipital areas, and both emotions with the hippocampus and midbrain [19]. However, it must be acknowledged that, since the distinction between shame and embarrassment is not sharp, being classified according to the private-public, moral-conventional, or low-high intensity dimensions, it is not easy to establish which brain areas are involved in processing these emotions, and which areas might selectively process one of the two emotions.

If the difference between shame and embarrassment is not as clear-cut, the distinction between guilt and the other two emotions is more evident. Guilt occurs when the violation of social norms induces harm or suffering to other individuals [3,4,37], typically in a relationship or among members of the same group [38]. Differently from shame and embarrassment, in which the self and the persona, respectively, are perceived as defective, in guilt, a specific action is typically perceived as wrong [34,37,39]. The occurrence of guilt induces remorse and behavioral responses that aim to repair the wrong action [18]. This difference in the focus of shame and guilt, self-oriented and other-oriented, respectively, has important consequences on empathy for other people: while guilt tends to increase the empathic concern towards other people, empathic responses seem to be disrupted by the self-oriented distress associated with shame [18].

In a review, Bastin and collaborators [19] suggested that guilt processing was selectively associated with the ventral anterior cingulate cortex (ACC), the precuneus, and premotor and posterior temporal areas. In addition, both guilt and shame processing were associated with the anterior insula and the dorsal ACC, and both guilt and embarrassment processing were associated with the dorsomedial prefrontal cortex (dmPFC), the vlPFC, and the anterior temporal lobe (ATL) [19]. Moreover, a recent meta-analysis [40] partially confirmed the guilt neural substrates proposed by Bastin and collaborators (2014), reporting the activation of the precuneus, the dorsal ACC, the dmPFC, and posterior temporal areas in association with guilt processing [40].

However, it is worth noting that studies investigating self-conscious emotions used heterogeneous methods that prevent any firm conclusions from being drawn. For this reason, we have run a meta-analysis study including neuroimaging research on the neural substrates of negative self-conscious emotions, i.e., to pinpoint brain areas consistently associated with shame/embarrassment and guilt processing. Since the distinction between shame and embarrassment is still a matter of debate, and since some of the studies in the literature did not distinguish clearly between the two emotions, we decided to include shame and embarrassment in one category. We predicted that shame/embarrassment and guilt may show different brain activations mirroring behavioural differences related to the emotions, together with some shared activations in light of their moral-self-conscious nature.

## 2. Materials and Methods

In order to find studies investigating the neural underpinnings of shame, embarrassment, and guilt, we conducted a research on PubMed (URL = https://www.ncbi.nlm.nih.gov/pubmed/ accessed on 1 January 2022) using the terms ((“fMRI” OR “functional magnetic resonance imaging” OR “PET”) AND (“shame” OR “embarrassment” OR “guilt” OR “moral emotions” OR “self-conscious emotions” OR “moral violations” OR “social standard violation”)) and setting a range of dates between 1 January 1995 and 13 February 2023. This research identified 169 studies.

Subsequently, we refined our research by applying the following criteria:
(1)Papers originally published in English;(2)fMRI or PET studies including task-related whole brain analyses. Studies reporting a region of interest (ROI analyses, resting-state fMRI analyses, diffusion tensor imaging (DTI), or voxel-based morphometry (VBM)) were excluded;(3)Participants were healthy adults: in the case of studies involving neurological or psychiatric patients, children, or adolescents, we considered only contrasts involving healthy controls, if reported;(4)Studies investigating the neural underpinnings of shame and guilt were included into two different sets, for two distinct meta-analyses. Specifically, we included studies contrasting shame/embarrassment vs. neutral or other emotional conditions, and guilt vs. neutral or other emotional conditions. Studies failing to distinguish embarrassment/shame and guilt were excluded.

Since the difference between shame and embarrassment is not clear-cut, as they can be classified according to different criteria, since they elicit the same physiological reactions and the same action tendencies, and since their distinction is still a matter of debate, we decided to include both shame and embarrassment in the same set.

This method allowed us to identify 18 studies for the shame/embarrassment set (188 foci, 439 total subjects) and 22 studies (one study was excluded in both sets because the contrast of interest did not show any significant result) for the guilt set (134 foci, 571 total subjects) (see Table 2). The most used paradigm in the studies analysed was emotion induction through verbal scripts (shame/embarrassment = 6, guilt = 8), pictures (shame/embarrassment = 5), both scripts and pictures (guilt = 3), vignettes (shame/embarrassment = 5), or movies (guilt = 1), while a few studies used the recollection of autobiographical memories through verbal scripts (shame/embarrassment = 1; guilt = 3), interpersonal games (shame/embarrassment = 1, guilt = 5), obedience paradigm (guilt = 1), or implicit association task (guilt = 1).

### Statistical Analysis

Analyses were conducted using the software GingerALE v3.0.2 (URL = http://brainmap.org/ accessed on 13 February 2023). The activation likelihood estimation method, implemented in the software [75,76,77], uses probability theory to define the spatial convergence of foci reported in the selected studies. Specifically, a Gaussian blur with an empirically-derived full width half maximum (dependent on the number of participants included in the study) is applied to each focus from a single study. Then, all the foci from a single study are represented in a modelled activation map and voxel-wise ALE scores are computed combining all the individual maps. To distinguish between true convergence of foci from random noise, a permutation test is applied. We adopted the method described by Turkeltaub et al. [77] that minimises within-study effects, preventing the summation of foci from the same experiment that are placed close to each other. For studies reporting between-subjects contrasts, we used the number of participants included in the smallest group as the total number of study participants.

The analyses were performed on the studies’ coordinates in MNI space. So, in the case of studies reporting coordinates in Talairach space, we converted them to MNI space using the coordinate converter of the GingerALE software, while we kept the same coordinates in studies reporting results in MNI space. For each set of studies, we performed the meta-analysis applying a cluster-level family-wise error correction using an uncorrected *p*-value < 0.001 for individual voxels, 1000 permutations, and a cluster-level threshold of *p* < 0.05, as suggested by Eickhoff and collaborators [78]. Finally, we performed further analyses to show possible overlaps or differences among the two emotions. We ran (1) a conjunction analysis aiming to elucidate common neural activations of shame/embarrassment and guilt; and (2) contrast analyses in order to highlight specific neural activations of either shame/embarrassment or guilt. Contrast analyses were performed subtracting one of the outputs of the previous analyses (ALE images) to the other (i.e., Shame/Embarrassment vs. Guilt, Guilt vs. Shame/Embarrassment). Then, simulations on data created by pooling the original data of both study groups into two new groups (same sample size as the original groups) were performed. Subsequently, a new subtraction map was computed with the two new datasets (subtracting one to the other) and it was compared to the true data. After 1000 permutations, a voxel-wise *p*-value image revealed, for each voxel, where the real data is located in the distribution of all the possible values (for that specific voxel). Values were converted into z-scores. We adopted an FDR correction (FDR pN in GingerALE) with *p* < 0.05 [76,79,80]. The specific contribution of each study to cluster formation can be found in the Appendix A. Results are visualised using SurfICE (URL = https://www.nitrc.org/projects/surfice/ accessed on 1 March 2023).

## 3. Results

### 3.1. Shame/Embarrassment

The meta-analysis on shame/embarrassment revealed five significant clusters (see Figure 1 and Table 3). One cluster included the left anterior insula and the pars orbitalis of the left inferior frontal gyrus (cluster 1), while three clusters were located within the frontal lobes and included the left anterior cingulate cortex (cluster 2), the right inferior frontal gyrus (IFG, cluster 3), and the right precentral gyrus (cluster 4). It is worth noting that clusters 3 and 4 received an important contribution from studies contrasting self- vs. other-face stimuli (see Appendix A). The other clusters were located within the medial portion of the left thalamus (cluster 5).

### 3.2. Guilt

The meta-analysis on guilt revealed two significant clusters (see Figure 2 and Table 4). One cluster was located at the level of the insula, involving also the orbital part of IFG (cluster 1). The other cluster was located on the posterior part of the left middle temporal gyrus, extending also to the angular gyrus at the junction between the temporal and parietal lobe (cluster 2).

### 3.3. Contrast Analyses

Conjunction analyses (see Figure 3 and Table 5) showed that both shame/embarrassment and guilt shared the activation of one cluster located within the left dorsal anterior insula and the pars orbitalis of the left inferior frontal gyrus. Contrast analyses (see Figure 3 and Table 5) revealed two significant clusters for the contrast ‘shame/embarrassment vs. guilt’, both at the level of the anterior cingulate cortex. The contrast ‘guilt vs. shame/embarrassment’ yielded no significant result.

## 4. Discussion

In the current meta-analysis, we analysed the functional neuroimaging literature on shame/embarrassment and guilt with the aim to identify the brain areas consistently associated with the processing of the two emotions. The results show that both emotions are associated with the activation of the left anterior insula, but they also show specific sets of areas involved in the processing of shame/embarrassment.

### 4.1. The Shame/Embarrassment Network

The occurrence of self-emotional distress in association with shame/embarrassment [4,18,81] might explain the association of the processing of these emotions with the dorsal ACC (cluster 2), the left anterior insula (cluster 1), and the medial nuclei of the thalami (cluster 5). Neuropsychological studies have highlighted that patients with dorsal ACC lesions, typically made in order to treat drug-resistant pain [82], are still able to perceive and correctly localise painful sensations, but such sensations are not distressing anymore [83]. Moreover, it is worth noting that the surgical lesion of the dorsal ACC also leads to a reduced concern about the opinions or the social judgement of other people [84], and can be used in the treatment of drug-resistant obsessive-compulsive disorder, a psychiatric syndrome which is often associated with extremely intense shame experiences [85]. Medial thalamic nuclei are thought to be involved in affective aspects of physical pain perception and attachment-related processes [86]. This set of areas is highly overlapping with those involved in the processing of both physical and social pain [87]. Social pain is the unpleasant experience associated with damage to social bonds or to social values (e.g., rejection, negative social evaluations, bereavement), and is thought to be processed by part of the neural circuit involved in processing physical pain [88]. Shame and embarrassment are thought to be important aspects of social pain, since they might signal that the social standards of others are not met [88].

The meta-analysis on shame/embarrassment also revealed clusters within the right premotor area (cluster 4) and right IFG (cluster 3), which have been associated with motor and speech inhibition [89,90], and which is consistent with the action tendencies associated with shame/embarrassment. Indeed, in contrast with guilt, which is often associated with pro-social behaviour aiming to repair the transgression that has occurred [18], shame and embarrassment lead to a reduction of social presence, speech, and movements [25,33], which could explain the activation of areas involved in motor and speech inhibition in shame/embarrassment processing. Hence, the presentation of shameful or embarrassing stimuli might automatically activate behavioral motor scripts aiming to reduce social presence. However, it is worth noting that studies contrasting self- vs. other-faces were important contributors to the generation of these clusters, as previous meta-analyses on neuroimaging studies on self-face recognition have reported [91,92]. Their relevance in the self-face processing network, especially for the right IFG, might be associated with the ability to differentiate self from other information [92].

### 4.2. The Guilt Network

Differently from shame and embarrassment, guilt is thought to be associated with social abilities, such as empathy and theory of mind, which were proposed to be specifically related to guilt generation [19,93]. In our meta-analysis, we found an association between guilt processing and TPJ, which was reliably found as a crucial area for distinguishing self- and other-actions, and representing other individuals’ mental and affective states, see [94] for a meta-analysis. Although the association between guilt and TPJ did not reach the significance level in the contrast analyses (i.e., guilt vs. shame/embarrassment), considering the convergence with a previous meta-analysis of guilt processing [40] and the associations between guilt processing and theory of mind, and theory of mind and TPJ, we believe that TPJ should be taken into consideration as a crucial area in the processing of guilt. However, the association between guilt, empathy, and theory of mind is not univocal. On the one hand, guilt is thought to increase the understanding of others’ affective and mental states [18]; on the other, taking others’ perspective and empathising with others seem to be crucial in order to experience guilt [95]. Hence, our results might refer to functions that are cause or consequence of the emotional experience.

### 4.3. Common Areas

The anterior insula was found in association with a wide variety of tasks, see [96]. Among the cognitive functions associated with the anterior insula that also include interoception, pain perception, and body awareness, it is worth mentioning its role in emotional awareness [96], arousal, and self-reflection, e.g., [97,98]. In addition, the lesion of this area is associated with pain asymbolia [99], a condition in which patients are still able to localise a painful stimulation and to identify it as pain but they lose all the unpleasant aspects (e.g., bodily, emotional, and behavioural signs) of pain [100]. The same type of patients showed reduced arousal ratings and an attenuated valence rating to emotional stimuli compared to both pathological and healthy controls [101]. The interpretation of these findings is not univocal. If, on the one hand, they might reflect the impairment in arousal processing, on the other they might also be caused by a deficit in emotional awareness. In addition, functional neuroimaging studies on healthy individuals investigating self-referential processing found anterior insula activation, e.g., [97,98]. Given that in our meta-analysis a high number of the studies included contrasted shame/embarrassment or guilt vs. neutral baselines, their common association with the anterior insula might simply reflect their arousal. Hence, the association between negative self-conscious emotion processing and the activation of the left anterior insula in functional neuroimaging studies might reflect their intensity, the awareness of the subjective experience of shame/embarrassment and guilt, or self-directed evaluation processes that are necessary in order to generate both guilt and shame experiences.

Our conjunction analysis did not show the involvement of the mPFC in representing both shame/embarrassment and guilt. Although our meta-analysis on shame/embarrassment revealed the activation of the ACC and the left mPFC (cluster 2), the same analysis on guilt did not show this cluster of activation. It is worth noting that these clusters of activation overlap with the results of a previous meta-analysis on guilt processing [40]. The mPFC represents a high-level integration area and is thought to support different aspects of social and affective processing [102,103] ranging from self-reflection [104], person perception [105], affective appraisal [106,107], theory of mind [108], learning, and predicting actions outcomes [109]. Moreover, the same area was found to be active in functional neuroimaging studies investigating moral judgement when moral evaluations were contrasted with non-moral or neutral baselines [110]. If, on the one hand, it has been proposed that the mPFC associates external stimuli (e.g., context-based information) with their socio-emotional value through a connection with anterior temporal lobes [8], on the other, it might be involved in self-referential processing (e.g., representation of traits, abilities, attitudes, and behaviours regarding the self), which is necessary in order to generate self-conscious emotions. This latter hypothesis seems to be confirmed by neuropsychological studies showing that patients with damage in the mPFC were impaired in self-referential memory [111]), self-evaluation [112], and self-referential verbal production [113].

## 5. Conclusions

Our meta-analysis revealed common and distinct neural substrates for the processing of shame/embarrassment and guilt. While the activation of the left anterior insula was associated with both shame/embarrassment and guilt processing, the pain network, including the medial thalami, the dorsal ACC, and premotor areas, was specifically associated with shame/embarrassment processing, while the left TPJ was associated with specific guilt processing.

## 6. Limitations

The main limitation of our work is the small number of studies investigating shame and embarrassment separately, which did not allow us to perform distinct meta-analyses on embarrassment and shame, as well as the relatively small number of participants included in most of the studies. The wide variety of paradigms investigating self-conscious emotions, including reading scripts, viewing vignettes, and recalling autobiographical memories, might affect the reliability of the results. Further studies investigating self-conscious emotions are necessary to better characterise common and specific brain networks involved in their processing.

## Figures and Tables

**Figure 1 brainsci-13-00559-f001:**
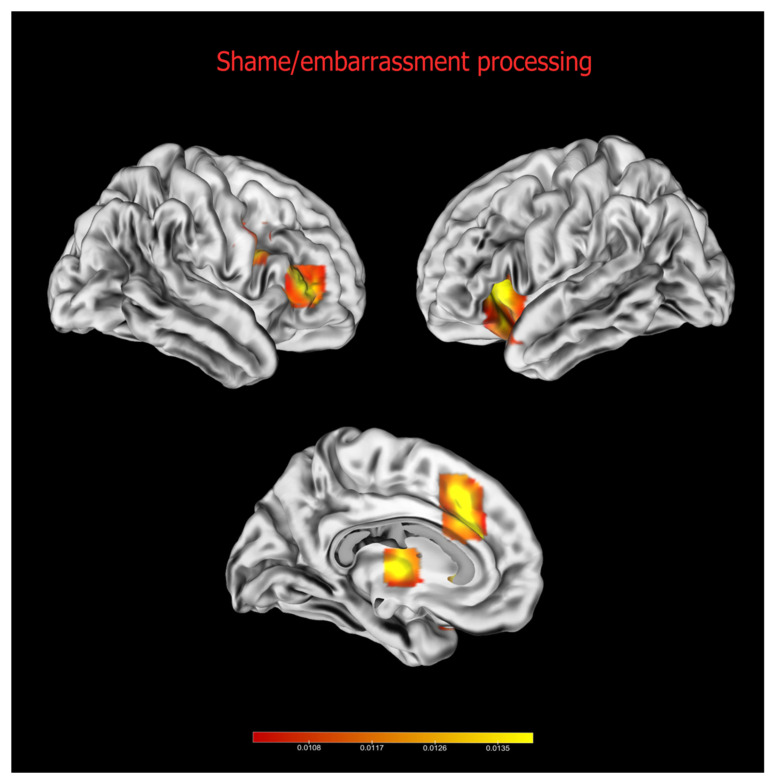
Results of the meta-analysis on shame/embarrassment neural correlates. Bars represent ALE scores.

**Figure 2 brainsci-13-00559-f002:**
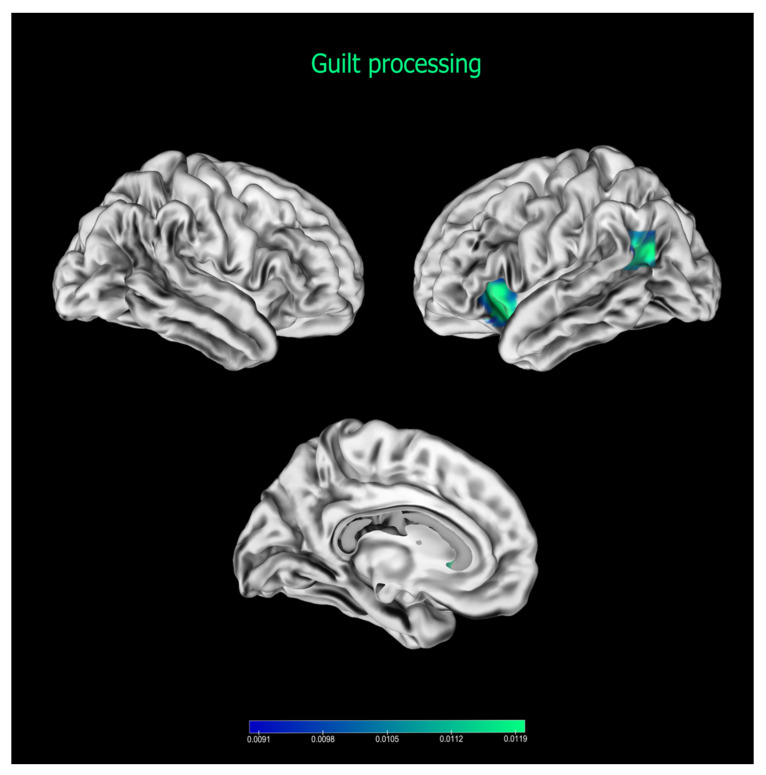
Results of the meta-analysis on guilt neural correlates. Bars represent ALE scores.

**Figure 3 brainsci-13-00559-f003:**
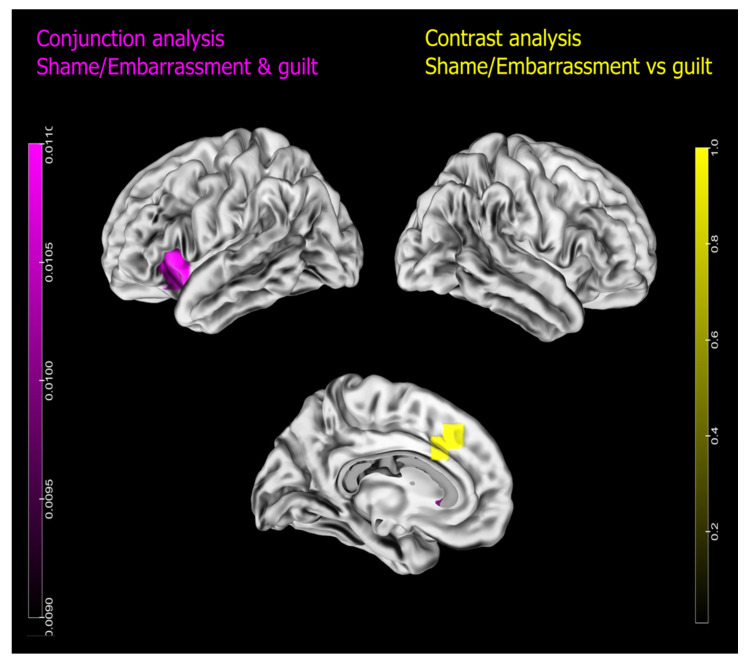
Contrast analysis results. In purple, conjunction analysis (shame/embarrassment and guilt); in yellow, specific activations of shame/embarrassment vs. guilt.

**Table 1 brainsci-13-00559-t001:** Differences between shame and guilt.

	Shame	Guilt
Target	What we are: related to the entire self. ”I’m bad”	What we do: related to specific behaviours. “What I did has been bad”
Level	Interpersonal—it occurs only with others	Intrapsychic—it occurs alone
Emotional activation	Painful	Less painful
Emotional perception	Difficult to recognize	Easy to recognize
Action tendency	Motivates hiding and inhibition	Motivates reparation of the situation
Relation with aggression, hostility, violence, externalization	Increased for shame-proneness individuals	Decreased for guilt-proneness individuals
Scapegoat	Blame mainly others	Blame myself
Responsibility	Deflected outward	Accepted

Adapted from Grecuccu et al., 2021 [4] and Tangney et al., 2007 [18].

**Table 2 brainsci-13-00559-t002:** Studies investigating shame/embarrassment and guilt brain processing.

Subset	Authors	Paradigm	Stimulus Type	Contrasts	Foci	Subjects (Females)
Shame/ embarrassment	Bas-Hogendam et al., 2017 [41]	Induction	Verbal scripts	Unintentional violations > neutral	5	21 (15)
	Bastin et al., 2021 [42]	Induction	Verbal scripts	Shame during reflection	None	42 (42)
	Berthoz et al., 2002 [43]	Induction	Verbal scripts	Unintentional violations > normal	15	12 (0)
	Finger et al., 2006 [44]	Induction	Verbal scripts	Moral and social with audience > social and neutral without audience	2	16 (-)
	Krach et al., 2011 [45]	Induction	Vignettes	Vicarious embarrassment > neutral	9	32 (17)
	Krach et al., 2015 [46]	Induction	Vignettes	Social pain > social neutral	17	16 (0)
	Laneri et al., 2017 [47]	Induction	Vignettes	Empathic embarrassment > neutral	14	51 (21)
	Melchers et al., 2015 [48]	Induction	Pictures	Vicarious embarrassment > neutral	6	60 (39)
	Mayer et al., 2020 [49]	Induction	Vignettes	([allo: shared > neutral + non-shared > neutral] ∩ [ego: shared > neutral + non-shared > neutral]).	13	48 (26)
	Michl et al., 2012 [50]	Induction	Verbal scripts	Shame > neutral	10	14 (7)
	Moll et al., 2008 [8]	Induction	Verbal scripts	Embarrassment > neutral	5	12 (6)
	Morita et al., 2008 [51]	Induction	Self- and other-faces	Self-face > other-face	9	19 (10)
	Morita et al., 2012 [52]	Induction	Self- and other-faces	Self-face > other-face	29	15 (2)
	Morita et al., 2013 [53]	Induction	Self- and other-faces	Self-face > other-face	17	32 (16)
	Morita et al., 2016 [54]	Induction	Self- and other-faces	Self-face > other-face	13	18 (0)
	Stroth et al., 2019 [55]	Induction	Vignettes	f-cg: (sks + nks) > ns	2	9 (9)
	Takahashi et al., 2004 [56]	Induction	Verbal scripts	Embarrassment > neutral	10	19 (9)
	Wagner et al., 2011 [57]	Recollection	Verbal scripts	Shame > neutral	10	15 (15)
	Zhu et al., 2018 [58]	Interpersonal game	Pictorial stimuli (dots)	Shame > happiness	2	30 (17)
Guilt	Basile et al., 2010 [59]	Induction	Verbal and facial stimuli	Guilt > anger and sadness	3	22 (13)
	Bastin et al., 2021 [42]	Induction	Verbal scripts	Guilt during reflection	None	42 (42)
	Cheng et al., 2021 [60]	Obedience paradigm	Movie	Positive correlation of guilt ratings in the harm > neutral condition	10	61 (32)
	Dominguez et al., 2018 [61]	Induction	Pictures	Incorrect versus correct shooting decisions	2	48 (35)
	Finger et al., 2006 [44]	Induction	Verbal scripts	Moral > social and neutral	5	16 (-)
	Fourie et al., 2014 [62]	Implicit association task	Verbal and facial stimuli	Prejudice feedback > neutral feedback	5	22 (22)
	Gilead et al., 2016 [63]	Induction	Verbal scripts	Guilt > anger, joy, pride	10	19 (14)
	Gradin et al., 2016 [64]	Interpersonal game	Verbal	Defection > cooperation	6	25 (17)
	Green et al., 2012 [65]	Induction	Verbal scripts	Guilt > indignation (within HC)	7	22 (18)
	Kedia et al., 2008 [66]	Induction	Verbal scripts	Guilt > self-anger	4	29 (14)
	Li et al., 2020 [67]	Interpersonal game	Pictorial stimuli (dots)	Out-group_ Commit > Out-group_ Observe	2	31 (19)
	Michl et al., 2012 [50]	Induction	Verbal scripts	Guilt > neutral	19	14 (7)
	Moll et al., 2008 [8]	Induction	Verbal scripts	Guilt > neutral	6	12 (6)
	Molenberghs et al., 2015 [68]	Induction	Video	Civilians > Soldiers	3	48 (24)
	Morey et al., 2012 [69]	Induction	Verbal scripts	Positive correlation of guilt	6	16 (0)
	Nihonsugi et al., 2021 [70]	Interpersonal game	Pictorial stimuli (cross)		6	52 (26)
	Peth et al., 2015 [71]	Recollection	Verbal	Guilty action > neutral	10	20 (6)
	Shin et al., 2000 [72]	Recollection	Verbal scripts	Guilt > neutral	8	8 (0)
	Takahashi et al., 2004 [56]	Induction	Verbal scripts	Guilt > neutral	5	19 (9)
	Ty et al., 2017 [73]	Induction	Verbal and pictorial stimuli	Restitution > harm	1	18 (9)
	Wagner et al., 2011 [57]	Recollection	Verbal scripts	Guilt > neutral	10	15 (15)
	Yu et al., 2013 [74]	Interpersonal game	Pictorial stimuli (dots)	Self-incorrect > both incorrect	1	24 (11)
	Zhu et al., 2019 [58]	Interpersonal game	Pictorial stimuli (dots)	Guilt > happiness	5	30 (17)

**Table 3 brainsci-13-00559-t003:** Results of the meta-analysis on shame/embarrassment processing.

Cluster	Volume (mm^3^)	Coordinates			ALE Value (^10^3^)	Lateralisation	Anatomical Label	BA
		x	y	z				
1	3592	−32	24	0	25.70	Left	Insula	13
		−38	20	−18	15.48		IFGorb	47
2	1984	−10	28	28	21.26	Left	Anterior cingulate gyrus	32
		−6	24	40	20.42		Medial frontal gyrus	8
3	1008	48	36	6	24.95	Right	IFGtri	46
4	840	50	8	28	19.80	Right	Precentral gyrus	9/6
5	760	−4	−8	6	21.64	Left	Thalamus	

Note: the table shows the results of the meta-analysis on shame/embarrassment neural correlates. BA = Brodmann’s area, IFGorb = inferior frontal gyrus *pars orbitalis*, IFGtri = inferior frontal gyrus *pars triangularis*. Coordinates reported are in MNI space.

**Table 4 brainsci-13-00559-t004:** Results of the meta-analysis on guilt processing.

Cluster	Cluster Size (mm^3^)	Coordinates			ALE Value (^10^3^)	Lateralisation	Anatomical Label	BA
		x	y	z				
1	1656	−32	18	−10	24.88	Left	Insula/IFGorb	13
2	904	−46	−58	20	18.78	Left	STG/angular gyrus	18

Note: the table shows the results of the meta-analysis on guilt neural correlates. BA = Brodmann’s area, IFGorb = inferior frontal gyrus pars orbitalis, STG = superior temporal gyrus. Coordinates are in MNI space.

**Table 5 brainsci-13-00559-t005:** Contrast analyses results.

Cluster	Cluster Size (mm^3^)	Coordinates			Lateralisation	Anatomical Label	BA
		x	y	z			
Conjunction analysis							
1	1000	−36	20	−6	Left	Insula/IFGorb	13
Contrast analysis—Shame/Emb vs. Guilt							
1	80	−6	29	40	Left	Anterior cingulate cortex	
2	32	−5	20	33	Left	Anterior cingulate cortex	
Contrast analysis—Guilt vs. Shame/Emb							
No significant results							

Note: the table shows the results of contrast analyses. BA = Brodmann’s area, IFGorb = inferior frontal gyrus pars orbitalis. Coordinates are reported in MNI space.

## Data Availability

Data are available upon request to the corresponding author.

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
