# Peer review of "The Neural Signatures of Shame, Embarrassment, and Guilt: A Voxel-Based Meta-Analysis on Functional Neuroimaging Studies"

_brainsci, 2023, doi:10.3390/brainsci13040559_

Round 1
Reviewer 1 Report
The manuscript entitle "The neural signatures of shame, embarrassment and guilt: a voxel-based meta-analysis on functional neuroimaging studies" has been carefully reviewed. This article has significant content and overall writing is good. Abstract is concise and written all the information. Enough literature presented in the introduction. Results are clearly discussed. The figures and table are well presented in the manuscript. Overall, I recommend to accept the current form.
Thank you.
Dr.Muthuraju Sangu
Author Response
We would like to thank the reviewer for his/her appreciations
Reviewer 2 Report
This study examined common and distinct neural substrates for the processing of shame/embarrassment and guilt by using a meta analysis of fMRI studies. There seems to be some concern about the selection of studies to be included in the meta-analysis and the results.
Introduction:
In this meta-analysis, the authors included shame and embarrassment in the same category (lines 160). Nevertheless, in the last sentence of the introduction, the authors mention that "We predicted that shame, embarrassment and guilt may show different brain activations." It would be better to state in the introduction that shame and embarrassment are included in the same category and examine if shame/embarrassment and guilt show different brain activations.
Methods:
Why an fMRI study which directly compared guilt with shame (Takahashi et al. 2004) is not included in the meta analysis?
Results:
Cluster numbers in the manuscript should be corresponding to the cluster numbers in Tables (3 & 4). The authors report a cluster observed in the right fusiform gyrus as cluster 6 (line 222), but it is not shown in Table 3. There is a sensence “The same analysis on studies contrasting shame/embarrassment with a neutral baseline revealed only a cluster located on bilateral lingual gyri, corresponding to cluster 4 in the previous analysis”( lines 224-225) , but I cannot find the lingual gyri cluster in Table 3.
Also, why did the authors perform the additional analysis on studies contrasting shame/embarrassment with a neutral baseline? What is the purpose?
In the footnote of Table 3, the authors mention that some clusters did not reach the significance level when studies using facial stimuli are excluded from the analysis.
This analysis and its results cannot be ignored and need to be considered more carefully. Since previous studies using face stimuli have compared the self-face condition with the other-face condition, it is likely that brain regions showing self-face-specific activity were included in addition to brain regions involved in emotional processing of shame/embarrassment. Indeed, the right middle frontal gyrus (cluster 7 in Table 3) and the right precentral gyrus (cluster 8 in Table 3) have repeatedly been reported to be very important regions for self-face perception (see meta analysis: Hu et al. 2016; doi: 10.1016/j.neubiorev.2015.12.003.).
I think the results of this additional analysis should be clearly stated in the Result section along with the rationale, or the studies using facial stimuli should be excluded from the meta-analysis from the beginning.
The same concern may be relevant to the subtraction analysis (Table 5). If the studies using facial stimuli are included in the meta-analysis, it would be better to check the results when these are excluded from the analysis.
In Figure 1, a color bar (for green) is needed.
Discussion:
I think there is a problem with the interpretation of the right premotor area and the right middle frontal gyrus (which the authors refer to as "dlPFC" in the discussion) identified in the meta-analysis of studies on shame/embarrassment (around lines 344 and 354). The authors interpret that the former is related to motor and speech inhibition and the latter is related to top-down regulatory processes. However, as I pointed out earlier, these areas are core regions which are associated with self-face recognition. The disappearance of these activations when studies using self-face images are excluded from the analysis would be consistent with this view.
In the Discussion, the authors mention that “In our meta-analysis we found the association between guilt processing and TPJ… (lines 369-370)”, but the cluster found in the meta-analysis on guilt processing (Table 4, Figure 1) appears to be in the superior temporal gyrus, not in the TPJ. The author should be more careful in the discussion of the guilt-related area.
In the last part of the Discussion, (also in the footnote of Table 4), the authors mention that a cluster did not reach the significance level when studies using autobiographical memory recall tasks are excluded from the analysis. The authors should provide a clear rationale as to why such additional analysis was performed. More explanation is needed.
Minor points:
Please check the year of the cited papers.
Michl et al. 2012 ->2014
Morita et al. 2013 ->2014
Author Response
Comments and Suggestions for Authors
This study examined common and distinct neural substrates for the processing of shame/embarrassment and guilt by using a meta analysis of fMRI studies. There seems to be some concern about the selection of studies to be included in the meta-analysis and the results.
We thank the reviewer for the suggestions that significantly improved the quality of our manuscript.
Q1: Introduction:In this meta-analysis, the authors included shame and embarrassment in the same category (lines 160). Nevertheless, in the last sentence of the introduction, the authors mention that "We predicted that shame, embarrassment and guilt may show different brain activations." It would be better to state in the introduction that shame and embarrassment are included in the same category and examine if shame/embarrassment and guilt show different brain activations.
A1: We agree with the reviewer about the importance of mentioning in the introduction the inclusion of shame and embarrassment in the same category. Please see below:
[p.4] For this reason, we run a meta-analysis study including neuroimaging research on the neural substrates of negative self-conscious emotions, i.e., to pinpoint brain areas consistently associated with shame/embarrassment and guilt processing. Since the distinction between shame and embarrassment is still a matter of debate, and since some of the studies in literature did not distinguish clearly between the two emotions, we decided to include shame and embarrassment in one category. We predicted that shame/embarrassment and guilt may show different brain activations mirroring behavioural differences related to the emotions, together with some shared activations in light of their moral-self-conscious nature.
Q2: Methods: Why an fMRI study which directly compared guilt with shame (Takahashi et al. 2004) is not included in the meta analysis?
A2: Since another reviewer suggested to update the meta-analysis including in the database also more recent studies, we performed another study research with the same keywords and new analyses. In the new analyses Takahashi et al. 2004 was also included.
Results:
Q3: Cluster numbers in the manuscript should be corresponding to the cluster numbers in Tables (3 & 4). The authors report a cluster observed in the right fusiform gyrus as cluster 6 (line 222), but it is not shown in Table 3. There is a sensence “The same analysis on studies contrasting shame/embarrassment with a neutral baseline revealed only a cluster located on bilateral lingual gyri, corresponding to cluster 4 in the previous analysis”( lines 224-225) , but I cannot find the lingual gyri cluster in Table 3.
A3: We apologize for the errors reported. We have now checked the whole manuscript for inconsistencies among text and tables.
Q4: Also, why did the authors perform the additional analysis on studies contrasting shame/embarrassment with a neutral baseline? What is the purpose?
A4: The purpose of the additional analyses was only to obtain a more homogeneous sample for the analysis. However, since further analyses using neutral baselines decreased the number of studies included and, consequently, the power of the analyses, and since these additional analyses were not relevant from a methodological and theoretical points of view, we decided to not perform them again in the current version of the manuscript.
Q5: In the footnote of Table 3, the authors mention that some clusters did not reach the significance level when studies using facial stimuli are excluded from the analysis. This analysis and its results cannot be ignored and need to be considered more carefully. Since previous studies using face stimuli have compared the self-face condition with the other-face condition, it is likely that brain regions showing self-face-specific activity were included in addition to brain regions involved in emotional processing of shame/embarrassment. Indeed, the right middle frontal gyrus (cluster 7 in Table 3) and the right precentral gyrus (cluster 8 in Table 3) have repeatedly been reported to be very important regions for self-face perception (see meta analysis: Hu et al. 2016; doi: 10.1016/j.neubiorev.2015.12.003.). I think the results of this additional analysis should be clearly stated in the Result section along with the rationale, or the studies using facial stimuli should be excluded from the meta-analysis from the beginning. The same concern may be relevant to the subtraction analysis (Table 5). If the studies using facial stimuli are included in the meta-analysis, it would be better to check the results when these are excluded from the analysis.
A5: We decided to adopt a more conservative approach, reporting only results corrected for multiple comparisons and reaching the significance threshold. As we explained also for Q4 and Q9, we did not perform additional analyses in the current version of the manuscript, since they reduce the power of the analyses (as the number of studies include decreases). We decided to add the contribution of each study for cluster formation in the supplementary materials.
As it can be seen in the supplementary materials, Cluster 4 (Precentral gyrus) was sustained only by studies involving self-face stimuli and that Cluster 3 received great contribution by the studies by Morita, using self-facial stimuli, but also by the study by Melchers et al 2015 (using film clips). This was commented in the discussion.
[pp.11-12] However, it is worth noting that studies contrasting self- vs. other- faces gave an important contribution in the generation of these clusters, as previous meta-analyses on neuroimaging studies on self-face recognition reported (Hu et al., 2016; Platek et al., 2008). Their relevance in the self-face processing network, especially for right IFG, might be associated with the ability to differentiate self from other information (Platek et al., 2008).
Q6: In Figure 1, a color bar (for green) is needed.
A6: We modified the pictures, adding the appropriate colorbars
Discussion:
Q7: I think there is a problem with the interpretation of the right premotor area and the right middle frontal gyrus (which the authors refer to as "dlPFC" in the discussion) identified in the meta-analysis of studies on shame/embarrassment (around lines 344 and 354). The authors interpret that the former is related to motor and speech inhibition and the latter is related to top-down regulatory processes. However, as I pointed out earlier, these areas are core regions which are associated with self-face recognition. The disappearance of these activations when studies using self-face images are excluded from the analysis would be consistent with this view.
A7: We have now discussed the updated results, also in the light of the suggestion given by the reviewer, considering in particular the possible inclusion in the self-face processing network of Clusters 3 and 4.
[p. 7] It is worth noting that cluster 3 and 4 received an important contribution by studies contrasting self- vs. other- face stimuli (See supplementary materials).
[pp.11-12] However, it is worth noting that studies contrasting self- vs. other- faces gave an important contribution in the generation of these clusters, as previous meta-analyses on neuroimaging studies on self-face recognition reported (Hu et al., 2016; Platek et al., 2008). Their relevance in the self-face processing network, especially for right IFG, might be associated with the ability to differentiate self from other information (Platek et al., 2008).
Q8: In the Discussion, the authors mention that “In our meta-analysis we found the association between guilt processing and TPJ… (lines 369-370)”, but the cluster found in the meta-analysis on guilt processing (Table 4, Figure 1) appears to be in the superior temporal gyrus, not in the TPJ. The author should be more careful in the discussion of the guilt-related area.
A8: The cluster found in the meta-analysis on guilt processing encompasses superior temporal gyrus and angular gyrus, matching the area reported in the meta-analysis by Schurtz et al. (2017). In the current version of the manuscript, we modified the picture so that the extension of cluster 2 towards the parietal lobule is visible.
Schurz, M., Tholen, M. G., Perner, J., Mars, R. B., & Sallet, J. (2017). Specifying the brain anatomy underlying temporo‐parietal junction activations for theory of mind: A review using probabilistic atlases from different imaging modalities. Human brain mapping, 38(9), 4788-4805.
Q9: In the last part of the Discussion, (also in the footnote of Table 4), the authors mention that a cluster did not reach the significance level when studies using autobiographical memory recall tasks are excluded from the analysis. The authors should provide a clear rationale as to why such additional analysis was performed. More explanation is needed.
A9: In the current version of the manuscript we added in the database more recent studies, increasing the power of the analysis. We decided to not implement additional analyses in the new version of the manuscript, as the exclusion of studies using autobiographical memory recall, because:
- our results did not involve memory-related brain networks
- reducing the numerosity of the database decreased the power of the analyses
- we added a table with specific study contributions for each cluster, so that results are more transparent and it becomes easier to verify whether a specific cluster is associated with studies using similar paradigms/stimuli.
Minor points:
Q10: Please check the year of the cited papers.
Michl et al. 2012 ->2014
Morita et al. 2013 ->2014
A11: We corrected the mistakes in the references suggested by the reviewer.
Author Response
We thank the reviewer for the useful comments that improved the quality of our manuscript.
In the current manuscript, Piretti and colleagues present a meta-analysis of neuroimaging
studies investigating three self-conscious emotions: shame, embarrassment, and guilt. The authors have scrutinized the literature published until the end of 2018, retrieving a total of 15 articles concerning shame/embarrassment (pooled together for theoretical and statistical reasons), and 17 about guilt. Then, the authors extracted activation coordinates reported in each study and fed these data into an activation likelihood estimation model. They conducted a total of four analyses aimed at revealing (1) regions meta-analytically activated by the induction of shame/embarrassment, (2) brain areas meta-analytically activated by the induction of guilt, (3) regions common to the three self-conscious emotions, and (4) those selectively involved in one emotion but not the other (i.e., the “shame/embarrassment ≠ guilt” meta-analytic contrast). Piretti and coauthors report that both guilt and shame/embarrassment relate to activations of the left anterior insula. In addition, while the induction of guilt specifically activates left TPJ, shame/embarrassment is associated with the recruitment of the dACC, the dlPFC, the insula, and the thalamus.
I read with interest and care the manuscript by Piretti and colleagues, which I found very
interesting and timely. Overall the article is well written and presents a topic often neglected in the affective literature: the brain correlates of (so-called) non-basic emotions.
Although my impression of this work is very positive, I necessitate clarifications regarding some analytical choices, and I have some suggestions that the authors may want to incorporate in a revised version of the manuscript.
Major points:
Q1: Firstly, the literature search is updated to December 2018. I have rapidly searched the PubMed database using queries similar to those reported in the current article and found (at least) two articles that could be added to the database (e.g., Yu et al., 2020; Bastin et al., 2021). Therefore, I encourage the authors to update their literature search and re-run all analyses.
A1: We agree with the reviewer that updating the analyses included new studies could increase the quality of the paper. Hence, we updated our database including more recent papers and performed new analyses. The updated analyses yielded different results than those reported in the previous version of the manuscript, which were reported and discussed accordingly.
Please, see below.
[p.4] In order to find studies investigating the neural underpinnings of shame, embarrassment and guilt we conducted a research on PubMed (https://www.ncbi.nlm.nih.gov/pubmed/) using the terms ((“fMRI” OR “functional magnetic resonance imaging” OR “PET”) AND (“shame” OR “embarrassment” OR “guilt” OR “moral emotions” OR “self-conscious emotions” OR "moral violations" OR "social standard violation")), and setting a range of dates between January 1st 1995 and February 13th 2023. This research identified 169 studies.
Q2: Secondly, I was wondering why the authors opted for Talairach coordinates in their
meta-analysis. As far as I know, the Talairach template is used less frequently than the MNI in modern neuroimaging studies. Because any conversion between one coordinate system to another comes with an error of approximation, a safer option would be to select the reference coordinate system depending on the articles’ template space. In other words, in case MNI is used more often than Talairach in studies on guilt, shame, and embarrassment, the authors should convert the Talairach activations into MNI coordinates and not vice versa.
A2: In this new version of the manuscript we performed the analysis using MNI space as reference space, since most of the studies performed analyses in this coordinate system. This allowed us to perform less coordinate conversions, minimising, as suggested by the reviewer, the approximation error.
[p.7] The analyses were performed on studies’ coordinates in MNI space. So, in case of studies reporting coordinates in Talaraich space, we converted them to MNI space, using the coordinate converter of the GingerALE software, while we kept the same coordinates in studies reporting results in MNI space.
Q3: Thirdly, it is not clear to me whether the authors employed the canonical GingerALE pipeline for the “shame/embarrassment ≠ guilt” meta-analytic contrast. In the manuscript, they refer to this contrast with the term “subtraction analysis”. However, the procedure they describe has left me dubious. Did the authors create the combined list of foci for shame/embarrassment and guilt before running the metanalytic contrast? This is the recommended procedure in the GingerALE manual (section: 2.3 “Contrast Analyses”). If not, they should consider following the software guidelines. Otherwise, please be more specific in the methods section when describing the procedure for the “shame/embarrassment ≠ guilt” meta-analytic contrast.
A3: We used the GingerAle pipeline to perform contrast analyses. We have now better explain this analysis in the methods section. Please see below.
[p.7] Finally, we performed further analyses to show possible overlaps or differences among the two emotions. We run 1) a conjunction analysis aiming to elucidate common neural activations of shame/embarrassment and guilt; 2) contrast analyses in order to highlight specific neural activations of either shame/embarrassment or guilt. Contrast analyses were performed subtracting one of the outputs of the previous analyses (ALE images) to the other (i.e., Shame/Embarrassment vs. Guilt, Guilt vs. Shame/Embarrassment). Then, simulations on data created by pooling the original data of both study groups into two new groups (same sample size as the original groups) were performed. Subsequently, a new subtraction map was computed with the two new datasets (subtracting one to the other) and was compared to the true data. After 1000 permutations, a voxelwise P-value image revealed, for each voxel, where the real data is located in the distribution of all the possible values (for that specific voxel). Values are converted into z-scores. We adopted an FDR correction (FDR pN in GingerALE), with p < .05. (Eickoff et al., 2012, Laird et al., 2005; Zmigrod et al., 2016). Results are visualised using MricroGL (https://www.mccauslandcenter.sc.edu/mricrogl).
Q4: Related to this, the statistical significance of the meta-analytic contrast seems to be based on104 permutations (line 208 in the current version of the manuscript). One-hundred permutations do not provide sufficient precision in estimating p-values. Is this a typo? If not, the authors should consider running at least 1,000 permutations as they did for other analyses.
A4: We apologise for the typing error. Actually, the analyses were performed on 5000 permutations.
[p.7] After 1000 permutations, a voxelwise P-value image reveals for each voxel, where the real data is located in the distribution of all the possible values (for that specific voxel).]
Q5: My last point with respect to the contrast analysis concerns the lack of correction for multiple comparisons. The authors report results based on the liberal p < 0.05 threshold without adjusting for the number of comparisons. To corroborate their choice, they cite Laird and colleagues paper (2005), in which it is reported - in fact - that FDR correction (p < 0.05 FDRc) should be preferred over the uncorrected threshold of p < 0.0001 (a less liberal value than the one used in the current manuscript). Similarly, Eickhoff and colleagues (2012) do not advocate the use of uncorrected thresholds. Instead, in a more recent work establishing guidelines for coordinate-based meta-analyses (Müller et al., 2018), the GingerALE group indicated that: “[...] a lack of control for multiple comparisons also comes with the concurrent downside of a potential contamination of the meta-analytic results (which in turn may strongly influence the future literature) by chance discoveries. Hence, in the majority of cases meta-analytic results should be reported following correction for multiple comparisons.” (pg. 156)
And that: “In general, for ALE meta-analyses (and possibly also other coordinate-based meta-analyses) cluster-level FWE correction seems to be the most reasonable approach, as it entails low susceptibility to false positives in terms of convergence (Eickhoff et al., 2016).” (pg. 157)
A5: As suggested by the reviewer, we adopted a more conservative threshold in the contrast analyses using FDR pN, with P < .05.
[p.7] We adopted an FDR correction (FDR pN in GingerALE), with p < .05. (Eickoff et al., 2012, Laird et al., 2005; Zmigrod et al., 2016).
Q6: In light of all this, I recommend the authors to report and discuss only results surviving the correction for multiple comparisons. In this regard, the data suggest that, while there is evidence of significant convergence in brain regions of Table 3 for shame/embarrassment and in brain areas of Table 4 for guilt, there are no significant differences in convergence when directly comparing the two emotions. In other words, no brain region is significantly more involved in (or selective for) shame/embarrassment than guilt. Of course, this could also be due by the small number of studies included in the meta-analysis, which in turn suggests that - if any - the differences between the two emotions are small, particularly when assessed using univariate neuroimaging techniques.
As far as I am concerned, this will not hamper the relevance of their manuscript at all.
A6: We have now corrected our tables, updating them with the new findings, and paying attention to the occurrence of possible discrepancies. Regions not surviving the statistical threshold (corrected of multiple comparisons) were not reported in the tables (and throughout the paper).
Minor points:
Q7: In Table 1, the authors list the potential differences between shame and guilt. Although I find this summary extremely useful and interesting, it is not clear whether it is based on a specific model of self-conscious emotions or rather it is something that the authors propose based on a review of the existing literature. In any case, it should be clarified.
Also, please note that Tables 1 and 2 are not referenced in the main manuscript.
“The same analysis on studies contrasting shame/embarrassment with a neutral baseline
revealed only a cluster located on bilateral lingual gyri, corresponding to cluster 4 in the
previous analysis.” (lines 224-226). This is not consistent with what is reported in Table 3 and in the previous paragraph, where cluster 4 is labeled as “left pre-SMA / left dACC” and “right precentral”, respectively. Similarly, in the text the authors report meta-analytic convergence for shame/embarrassment in the right fusiform gyrus. However, there is no ventral occipito-temporal region listed in Table 3. Also, in caption of Table 3, the authors report that clusters from 5 to 8 did not survive statistical thresholding after removing studies using face stimuli. From Table 2 it seems that studies employing face stimuli are also those not including a neutral condition. Thus I am uncertain about how to interpret the caption and the results reported in Table 3 in light of what is written in the manuscript and detailed in Table 2. Please clarify. “The other cluster was located within the occipital lobes, on the midline, at the level of lingualgyri (cluster 4).” (lines 242-243) In the table summarizing results of the meta-analysis on guilt processing (i.e., Table 4) there is no cluster 4. Please clarify. In addition, according to the caption of the same table, it seems that the authors have conducted additional analyses after removing studies based on recollection.
This is, however, not reported in the main manuscript.
A7: We apologise for the many mistakes that were present in the manuscript. The updated analyses revealed slightly different results that were reported in the text, in the tables and in the figure. We decided to not implement additional analyses (e.g. removing studies using facial stimuli) in the new version of the manuscript because they were not relevant from a methodological and theoretical point of view and because, decreasing the number of studies included, reduced the power of the analyses.
We also added the caption for the tables when missing.
Q8: “Hence, the association between negative self-conscious emotion processing and the activation of left anterior insula in functional neuroimaging studies might reflect the awareness of the subjective experience of shame/embarrassment and guilt, its intensity, or self-directed evaluation processes that are necessary in order to generate both guilt and shame experiences.” (lines 293-296)
An alternative explanation could be that meta-analytic convergence in the anterior insula relates to differences in arousal between shame/embarrassment and neutral affect. The authors correctly acknowledge the tight association between aINS activity and arousal in the context of pain research but it seems that when it comes to interpreting the meta-analytic results they favor the idea that aINS has a self-reflective function. Yet, as the majority of studies (11 out of 17) included here are based on the “shame > neutral” contrast, I doubt that one explanation should be preferred over the other and I suggest the authors to emphasize this aspect.
A8: Actually, we did not mean to express our favor for one hypothesis on the other in the explanation of why aINS might be associated with shame/embarrassment and guilt processing. We have now modified the paragraph of the discussion about insula, in order to better explain that all the three possible explanations - arousal, awareness of the subjective experience or self-reflection - might be valid
[p.11] Given that in our meta-analysis a high number of the studies included contrasted shame/embarrassment or guilt vs. neutral baselines, their common association with anterior insula might simply reflect their arousal. Hence, the association between negative self-conscious emotion processing and the activation of left anterior insula in functional neuroimaging studies might reflect their intensity, the awareness of the subjective experience of shame/embarrassment and guilt, or self-directed evaluation processes that are necessary in order to generate both guilt and shame experiences.
Q9: I found a few typos in the manuscript that should be corrected (amendments are suggested in red):
- “They provide an immediate feedback that promote inhibition or reinforcement of
behaviour based on their positive or valence (Tangney et al., 2007; Grecucci et al., 2021;
Piretti et al., 2020).” (lines 60-61)
- “[...] we analyzed the functional neuroimaging literature on shame/embarrassment and
guilt with the aim to identify the brain areas consistently associated with the processing
of either emotions.” (lines 273-275)
- “The results show that either emotions to be associated with left anterior insula, but they
also show specific sets of areas involved in the processing of shame/embarrassment
and guilt.” (lines 275-277)
A9: We have now corrected the sentences suggested by the reviewer
1 Self-conscious emotions are evoked by self-reflection and self-evaluation (Tangney et al., 2007) and occur when social norms, or agreed-upon social rules, are violated (Bastin et al., 2016), providing an immediate feedback that promotes inhibition or reinforcement of behaviour (Tangney et al., 2007; Grecucci et al., 2021; Piretti et al., 2020).
2 In the current meta-analysis we analysed the functional neuroimaging literature on shame/embarrassment and guilt with the aim to identify the brain areas consistently associated with the processing of the two emotions.
3 The results show that both emotions are associated with left anterior insula, but they also show specific sets of areas involved in the processing of shame/embarrassment and guilt.

Round 2
Reviewer 2 Report
The authors responded to most of the comments I had raised, but I still have only one comment.
In the abstract and conclusion, the authors stated that the anterior insula is associated with shame/embarrassment and guilt processing, and that the (inferior anterior) insula is specifically involved in shame/embarrassment. The interpretation of the insula seems to be ambiguous. Since the contrast 'shame/embarrassment vs. guilt' did not show significant difference in the inferior anterior insula, it would be better to assume that the anterior insula is involved in both shame/embarrassment and guilt processing.
I would recommend writing "…the pain network especially medial thalami and dorsal ACC were specifically associated with shame/embarrassment processing…".
Author Response
A1: We agree with the reviewer and as suggested have removed the specification of the anterior inferior insula, which could have been ambiguous.
Reviewer 3 Report
The authors have addressed all my concerns. Their work represents a timely and valuable contribution to the study of self-conscious emotions and I recommend acceptance in the present form.
Luca Cecchetti
Author Response
We thank the reviewer for the feedback.